# Random Allelic Expression in Inherited Retinal Disease Genes

**Collin J. Richards \* and Jose S. Pulido**

Wills Eye Hospital, Mid Atlantic Retina, Philadelphia, PA 19107, USA; jpulido@willseye.org
\* Correspondence: collin.richards@jefferson.edu

**Abstract:** Inherited retinal diseases (IRDs) are a significant contributor to visual loss in children and young adults, falling second only to diabetic retinopathy. Understanding the pathogenic mechanisms of IRDs remains paramount. Some autosomal genes exhibit random allelic expression (RAE), similar to X-chromosome inactivation. This study identifies RAE genes in IRDs. Genes in the Retinal Information Network were cross-referenced with the recent literature to identify expression profiles, RAE, or biallelic expression (BAE). Loss-of-function intolerance (LOFI) was determined by cross-referencing the existing literature. Molecular and biological pathways that are significantly enriched were evaluated using gene ontology. A total of 184 IRD-causing genes were evaluated. Of these, 31 (16.8%) genes exhibited RAE. LOFI was exhibited in 6/31 (19.4%) of the RAE genes and 18/153 (11.8%) of the BAE genes. Brain tissue exhibited BAE in 107/128 (83.6%) genes for both sexes. The molecular pathways significantly enriched among BAE genes were photoreceptor activity, tubulin binding, and nucleotide/ribonucleotide binding. The biologic pathways significantly enriched for RAE genes were equilibrioception, parallel actin filament bundle assembly, photoreceptor cell outer segment organization, and protein depalmitoylation. Allele-specific expression may be a mechanism underlying IRD phenotypic variability, with clonal populations of embryologic precursor cells exhibiting RAE. Brain tissue preferentially exhibited BAE, possibly due to selective pressures against RAE. Pathways critical for cellular and visual function were enriched in BAE, which may offer a survival benefit.

**Keywords:** inherited retinal disease; genetics; random allelic expression; monoallelic expression

## 1. Introduction

Inherited retinal diseases (IRDs) represent an important and variable group of eye diseases that cause significant visual impairment among those afflicted. IRDs are now listed among the most common causes of childhood blindness in high- and middle-income countries [1]. In England and Wales, IRDs have overtaken other forms of blinding eye disease as the most common cause of blindness among the working age population [2]. There are currently 282 unique genes responsible for IRDs that have been mapped and identified [3]. Recent recommendations from the Monaciano Symposium have listed "understanding the pathogenic mechanisms underlying retinal dystrophies" as the first priority in the effort to address the challenges posed by IRDs [4]. The current understanding of IRDs groups them by inheritance pattern (autosomal, X-linked, and mitochondrial) and by phenotypic category (RP, rod–cone dystrophy, cone or cone–rod dystrophy, macular dystrophy, Leber's congenital amaurosis, congenital stationary night blindness, and other more complex phenotypes with extraocular syndromic involvement) [4]. It has been previously assumed that there is equal intracellular expression of maternal and paternal alleles, except in cases of X-chromosome inactivation (XCI), olfactory receptors, clustered protocadherins, and canonical imprinted genes [5]. Another mechanism for gene expression was first discovered in 2007, where monoallelic expression in human lymphoblastoid cell lines was demonstrated [6]. This allelic expression mechanism also demonstrates clonality, where random allelic expression (RAE) is mitotically heritable from parent to daughter cells [5].

This mechanism of autosomal allelic regulation affects the gene dose and differential expression of heterozygous variants, similar in concept to the process of lyonization observed in X-linked genes [5]. Recent work has developed new methods to reveal RAE genes and has generated large, open-source datasets of RAE and biallelic expression (BAE) genes [7]. Our study used this dataset, in concert with other datasets [8] and publicly available genetics resources, including the Gene Ontology (GO) knowledgebase [9,10], to define the roles of RAE and BAE among IRDs.

## 2. Materials and Methods

Genes listed in the Retinal Information Network (https://sph.uth.edu/retnet/, accessed on 14 June 2023) were assessed using only mapped and identified genes for further analysis. *OPA1* and *MFN2* were included in the RetNet™ dataset and were analyzed [3]. Mutations in these genes, traditionally associated with optic atrophy, result in primary degeneration of retinal ganglion cells, with subsequent ascending atrophy of the optic nerve, and were therefore included [11,12]. These genes were cross-referenced with those genes termed "high-confidence random allelic expression (hc-RAE) and high-confidence biallelic expression (hc-BAE)" in Supplementary Table S2 of Kravitz et al., 2023 [7]. The threshold to define RAE and BAE was empirically derived to yield XCI-like RAE. Each gene's RAE population frequency was compared with the mean for all autosomal and XCI genes combined, yielding a threshold for all tissues of $Z \geq 0.74$, to indicate autosomal genes with RAE properties similar to XCI genes, while strong biallelic expression was defined as $Z \leq 0$, with $0 < Z < 0.74$ being undefined. The threshold for significant RAE genes in brain tissue was set at $Z > 0.11$ and that for body tissue was set at $Z > 0.86$. High-confidence RAE and BAE were those genes independently replicated in both female and male datasets. Full methodology for empiric derivation of Z-score cutoffs may be found in Kravitz et al., 2023 [7]. Tissue-specific RAE and BAE thresholds were defined using the same Z-score cutoff for body tissue and brain tissue. Loss-of-function intolerance was determined by cross-referencing our dataset with the dataset from Tanner et al., 2022 [8]. We evaluated the molecular and biological pathways that are significantly enriched using the GO knowledgebase (http://geneontology.org/, accessed on 7 July 2023) (http://pantherdb.org/, accessed on 7 July 2023) for both hc-BAE and hc-RAE genes [9,10]. Results between groups were analyzed using T-tests for continuous variables and Chi-squared tests for categorical variables. *p*-values less than 0.05 were used to indicate statistical significance.

## 3. Results

A total of 282 unique genes were pulled from RetNet™, of which 262 were autosomal genes. Cross-referencing this autosomal gene list with genes from Supplementary Table S2 of Kravitz et al., 2023 [7], yielded a total of 184 unique autosomal genes for evaluation. Thirty six genes (19.6%) were inherited in an autosomal dominant fashion, 129 (70.1%) were inherited in an autosomal recessive fashion, and 19/184 (10.3%) exhibited both AD and AR inheritance patterns. In total, 31 genes (16.8%) exhibited RAE in all tissues (Table 1), and 153 (83.2%) exhibited BAE in all tissues (Supplementary Table S1).

The average z-scores for all tissues in males was −0.07 (range: −0.92 to 3.90, standard deviation: 0.95) and in females was −0.12 (range: −0.87 to 3.86, standard deviation: 0.87); for body tissue in males was −0.13 (range: −0.87 to 3.91, standard deviation: 0.90) and in females was −0.16 (range: −0.80 to 3.61, standard deviation: 0.84); and for brain tissue in males was 0.02 (range: −0.60 to 11.17, standard deviation: 1.44) and in females was −0.15 (range: −0.72 to 7.33, standard deviation: 0.99). All expression profiles for "all tissues", "brain tissue", and "body tissue" were concordant between genders, by definition, as the criteria for "high-confidence RAE" were those genes that met the predefined threshold for RAE for both males and females. Discordance is defined as gene expression profiles that do not match between tissue types (e.g., BAE in all tissues but RAE in brain tissue). Comparing the concordance of expression profiles between brain tissue and all tissues revealed that 17/104 (16.3%) genes were discordant. There were significantly more genes

exhibiting BAE in brain tissue versus all tissues ($p < 0.001$) (Table 2); 11 genes exhibited RAE in all tissues and BAE in brain tissue, while 6 genes exhibited BAE in all tissues and RAE in brain tissue. There was 100% concordance of expression profiles between all tissues and body tissue.

**Table 1.** List of genes from RetNet that exhibit random allelic expression (RAE) in all tissue types. AR: autosomal recessive; AD: autosomal dominant.

| Gene | Chromosome | Disease | Inheritance |
|---|---|---|---|
| *NPHP4* | 1 | Syndromic/systemic diseases with retinopathy | AR |
| *ESPN* | 1 | Deafness alone or syndromic | Both AR and AD |
| *TIMP3* | 22 | Macular degeneration | AD |
| *MERTK* | 2 | Retinitis pigmentosa | AR |
| *CDHR1* | 10 | Cone or cone–rod dystrophy | AR |
| *PHYH* | 10 | Syndromic/systemic diseases with retinopathy | AR |
| *RCBTB1* | 13 | Other retinopathy | Both AR and AD |
| *EFEMP1* | 2 | Macular degeneration | AD |
| *WFS1* | 4 | Deafness alone or syndromic | Both AR and AD |
| *WHRN* | 9 | Deafness alone or syndromic | AR |
| *PITPNM3* | 17 | Cone or cone–rod dystrophy | AD |
| *CDH23* | 10 | Deafness alone or syndromic | AR |
| *ABHD12* | 20 | Syndromic/systemic diseases with retinopathy | AR |
| *CYP4V2* | 4 | Retinitis pigmentosa | AR |
| *PPT1* | 1 | Syndromic/systemic diseases with retinopathy | AR |
| *USH1C* | 11 | Deafness alone or syndromic | AR |
| *ASRGL1* | 11 | Other retinopathy | AR |
| *PRPH2* | 6 | Cone or cone–rod dystrophy | AD |
| *TUB* | 11 | Syndromic/systemic diseases with retinopathy | AR |
| *CTNNA1* | 5 | Macular degeneration | AD |
| *PROM1* | 4 | Cone or cone–rod dystrophy | Both AR and AD |
| *VCAN* | 5 | Ocular–retinal developmental disease | AD |
| *TSPAN12* | 7 | Other retinopathy | AD |
| *PCYT1A* | 3 | Syndromic/systemic diseases with retinopathy | AR |
| *ACO2* | 22 | Optic atrophy | AR |
| *CFH* | 1 | Macular degeneration | AR |
| *MFN2* | 1 | Optic atrophy | AD |
| *INPP5E* | 9 | Bardet–Biedl syndrome | AR |
| *JAG1* | 20 | Syndromic/systemic diseases with retinopathy | AD |
| *CNGA1* | 4 | Retinitis pigmentosa | AR |
| *PLA2G5* | 1 | Other retinopathy | AR |

**Table 2.** List of genes with discordant expression profile between all tissues and brain tissue. RAE: random allelic expression; BAE: biallelic expression.

| Discordant Genes | Expression Profile | |
|---|---|---|
| | **All Tissues** | **Brain** |
| *PROM1* | RAE | BAE |
| *VCAN* | RAE | BAE |
| *TSPAN12* | RAE | BAE |
| *PCYT1A* | RAE | BAE |
| *ACO2* | RAE | BAE |
| *CFH* | RAE | BAE |
| *MFN2* | RAE | BAE |
| *INPP5E* | RAE | BAE |
| *JAG1* | RAE | BAE |
| *CNGA1* | RAE | BAE |
| *PLA2G5* | RAE | BAE |
| *IFT81* | BAE | RAE |
| *PDZD7* | BAE | RAE |
| *CRB1* | BAE | RAE |
| *CEP250* | BAE | RAE |
| *SLC25A46* | BAE | RAE |
| *CERKL* | BAE | RAE |

Comparing inheritance patterns showed that 111/153 (72.5%) hc-BAE genes and 18/31 (58.1%) hc-RAE genes had AR inheritance, 27/153 (17.6%) hc-BAE genes and 9/31 (29%) hc-RAE genes had AD inheritance, and 15/153 (9.8%) hc-BAE and 4/31 (13%) hc-RAE genes had both AR and AD inheritance, yielding no significant difference in inheritance pattern (AD versus AR) for BAE versus RAE ($p = 0.257$). There was no difference in the proportion of genes exhibiting loss-of-function intolerance between BAE and RAE genes, as loss-of-function intolerance was observed in 6/31 (19.4%) of hc-RAE genes and 18/153 (11.8%) of hc-BAE genes ($p = 0.193$) (Table 3).

**Table 3.** Genes that exhibit loss-of-function intolerance, and the associated chromosome, disease, and expression profile for all tissues. RAE: random allelic expression; BAE: biallelic expression.

| Gene | Chromosome | Disease | Expression Profile, All Tissues |
|---|---|---|---|
| CDHR1 | 10 | Cone or cone–rod dystrophy | RAE |
| PITPNM3 | 17 | Cone or cone–rod dystrophy | RAE |
| PROM1 | 4 | Cone or cone–rod dystrophy | RAE |
| ACO2 | 22 | Optic atrophy | RAE |
| INPP5E | 9 | Bardet–Biedl syndrome | RAE |
| NPHP1 | 2 | Bardet–Biedl syndrome | BAE |
| IFT172 | 2 | Bardet–Biedl syndrome | BAE |
| CEP290 | 12 | Bardet–Biedl syndrome | BAE |
| UNC119 | 17 | Cone or cone–rod dystrophy | BAE |
| SLC7A14 | 3 | Retinitis pigmentosa | BAE |
| AGBL5 | 2 | Retinitis pigmentosa | BAE |
| RD3 | 1 | Leber congenital amaurosis | BAE |
| IMPG2 | 3 | Retinitis pigmentosa | BAE |
| IFT81 | 12 | Cone or cone–rod dystrophy | BAE |
| CERKL | 2 | Cone or cone–rod dystrophy | BAE |
| SPATA7 | 14 | Leber congenital amaurosis | BAE |
| POC1B | 12 | Cone or cone–rod dystrophy | BAE |
| OPA1 | 3 | Optic atrophy | BAE |
| SNRNP200 | 2 | Retinitis pigmentosa | BAE |
| PDE6C | 10 | Cone or cone–rod dystrophy | BAE |
| PANK2 | 20 | Syndromic/systemic diseases with retinopathy | BAE |
| PCDH15 | 10 | Deafness alone or syndromic | BAE |
| ITM2B | 13 | Other retinopathy | BAE |

The molecular pathways that were significantly enriched for hc-BAE genes include photoreceptor activity (fold enrichment 29.14, *p*-value < 0.001, false detection rate (FDR) FDR $5.25 \times 10^{-2}$), tubulin binding (fold enrichment 4.42, *p*-value < 0.001, FDR $5.71 \times 10^{-2}$), purine ribonucleotide binding (fold enrichment 2.2, *p*-value < 0.001, FDR $3.86 \times 10^{-2}$), ribonucleotide binding (fold enrichment 2.18, *p*-value < 0.001, FDR $3.32 \times 10^{-2}$), and purine nucleotide binding (fold enrichment 2.15, *p*-value < 0.001, FDR $3.55 \times 10^{-2}$). There were no significantly enriched molecular pathways for hc-RAE genes in our cohort. The biological pathway that was significantly enriched (>100 fold) for hc-BAE genes was pigment granule aggregation in the cell center (*p*-value < 0.001, FDR $3.16 \times 10^{-2}$). The biological pathways that were significantly enriched (>100 fold) for hc-RAE genes were equilibrioception (*p*-value < 0.001, FDR $3.01 \times 10^{-2}$), parallel actin filament bundle assembly (*p*-value < 0.001, FDR $2.94 \times 10^{-2}$), photoreceptor cell outer segment organization (*p*-value < 0.001, FDR $1.12 \times 10^{-3}$), and protein depalmitoylation (*p*-value < 0.001, FDR $4.93 \times 10^{-2}$). A full list of the biologic processes showing significant overrepresentation among hc-RAE and hc-BAE is included in Supplementary Table S3. There were 38 biologic processes significantly enriched for both hc-BAE and hc-RAE genes (Figure 1).

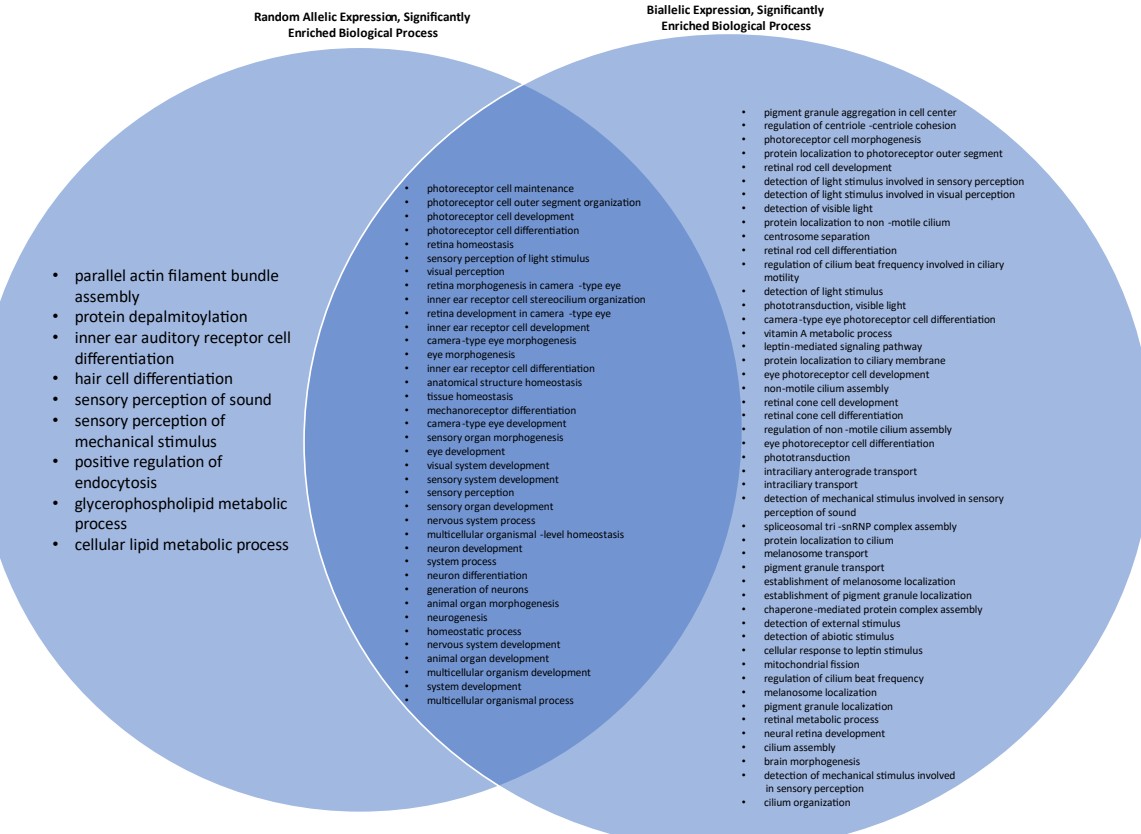

**Figure 1.** Venn diagram comparing significantly enriched biologic processes for genes that exhibit random allelic expression (RAE) and biallelic expression (BAE). Additional significantly enriched BAE processes seen in Supplementary Table S2.

We cross-referenced our dataset with the previously reported hc-RAE and hc-BAE genes located in significant topologically associated domains (TADs) from Kravitz et al., 2023, Supplementary Table S3 [7]. We found 3 TADs significantly enriched for hc-RAE genes (*CNGA1*, *CDH23*, and *ASRGL1*) and 10 TADs significantly enriched for hc-BAE genes (*HARS*, *ROM1*, *CABP4*, *BBS1*, *CDH3*, *CLN3*, *UNC119*, *OPA3*, *CEP250*, and *KIF3B*).

## 4. Discussion

Among genes that are known to be causative for IRDs, we identified 31 genes that exhibit RAE and 153 genes that exhibit BAE. These genes and their unique expression profiles may underly the characteristic phenotypes observed in many IRDs.

Allele-specific expression is an important component of phenotypic variability observed in genetic diseases [13–15]. It has been suggested that lyonization may be responsible for the characteristic fundus morphology observed within female carriers of X-linked recessive disease [16]. The morphogenesis of the RPE in vertebrates has been previously characterized and results in distinct clonal grouping patterns. In mammalian models, RPE development occurs in an "edge-biased" pattern, producing columns of clonal populations emanating peripherally from central progenitor RPE populations [17,18]. The central RPE populations maintain lower levels of mitotic activity and exhibit more cellular mixing in contrast to the peripheral populations [17]. This pattern of RPE development has similarly been observed in feline neurosensory retina development [19]. Mouse models have suggested that XCI occurs among clonal patches in a radial pattern, with alternating columns of *lacZ*-active and -inactive cells, suggesting that progenitor cell inactivation becomes fixed through subsequent generations [20]. The imprinting of allele expression profiles is more prevalent during embryologic development [13]. Furthermore, RPE development has been

shown to occur through two distinct morphological patterns: pinwheeling and spreading. Prospective RPE cells replicate via a pinwheel formation during optic vesicle elongation followed by spreading and elongation within columns during optic vesicle invagination. Adjacent tissue types (i.e., prospective posterior RPE cells and posterior retinal cells) move in a similar pattern, irrespective of final cell fate [21]. We speculate that genes exhibiting RAE may result in a similar phenomenon, with alternating columns of mutant and wild-type cells, emanating from a progenitor cell that was fixed in a location, resulting in radial columns of affected cells. For example, this may explain the appearance of radial drusen in autosomal dominant radial drusen (ADRD, also known as Doyne Honeycomb Dystrophy or Mallatia Levetinese).

Our study revealed discordant expression profiles between all tissues and brain tissue. There was a significant difference in the relative overexpression of BAE in brain tissue when compared to all tissues. The trend of reduced RAE in brain tissue suggests a selection pressure against RAE in brain tissue that could make these tissues more resistant to single gene mutations. Prior work has shown that BAE is enriched in genes for essential nuclear functions, mutation-intolerant genes, and is preserved evolutionarily [7,14]. Selection against RAE in brain tissue versus body tissue may reflect a difference in mutation tolerance for these tissues (i.e., body tissue may better tolerate a mutation that would otherwise portend a genetic disadvantage if occurring in brain tissue). Also, an evolutionary drive to preserve critical neurological pathways essential for survival may account for decreased RAE in brain tissue. Within XCI, significant intra-tissue and inter-tissue variability exists within individuals [22]. RAE may exhibit similar variability, a phenomenon that would not be captured in our evaluation of brain tissue from various regions. A potential outcome of this would be diseases with variable expressivity and diseases that display local, ocular phenotypes, without a systemic association.

Among our cohort, loss-of-function intolerance was not significantly different between hc-RAE and hc-BAE genes. This is in contrast to Kravitz et al. [7], who found that hc-RAE genes were significantly enriched for genes tolerant to homozygous loss-of-function variants. This discrepancy may reflect a true difference among genes responsible for IRDs or may be a result of inadequate sample size.

Our analysis of molecular pathways revealed that photoreceptor activity, tubulin binding, and purine ribonucleotide binding were molecular pathways significantly enriched by hc-BAE genes, while no molecular pathways were significantly enriched by hc-RAE genes. These pathways enriched by hc-BAE genes are crucial for vision and cellular function, suggesting that hc-BAE genes may be protective for critical molecular pathways and may offer a survival benefit. This follows prior work suggesting that diploid organisms have a survival benefit by nature of their ability to mask recessive, deleterious alleles [23]. RAE genes have been shown to be enriched in pathways of cell surface receptors and developmental regulators, which may drastically increase the ability of sensory cells to respond to diverse environmental stimuli and increase neuronal plasticity [24,25]. Our findings are in alignment with this trend, as RAE is enriched in many cell membrane-specific and sensory pathways, including protein depalmitoylation, the positive regulation of endocytosis, and parallel actin filament bundle assembly. Interestingly, there are many biological pathways in which both RAE and BAE genes are enriched, many of which are related to the development, differentiation, maintenance, and organization of the retina and other sensory systems.

There are limitations to this study. It used the data from RetNet to determine what genes are involved in IRD, and there are probably more genes involved that are still not known, and thus, not in that dataset. Additionally, it involved the use of data from the Kravitz dataset [7]. Thus far, the data from that dataset have been shown to be valid, but there could be assumptions that might invalidate the results from certain genes, but with the information that we have at this time, these results appear to be reasonable. Another important limitation is the lack of tissue-specific data for the retina. The brain tissue analyzed in the Kravitz dataset [7] is from the Genotype Tissue Expression (GTEx) v8 release, which

includes tissues from the thalamus and hypothalamus, among other brain regions [26]. The human retina is embryonically derived from the diencephalon, the precursor to the thalamus and hypothalamus [27,28]. As expression profile imprinting occurs significantly more during embryonic development [13], this study serves as a first approximation to understanding RAE in IRDs using data from tissue with a common embryonic precursor. It is known that RAE status is variable between different cell types [25], which may result in similar region-to-region variation in CNS function observed in XCI [22]. Future studies should investigate tissue-specific expression profiles for retinal tissue.

This study is the first to characterize the presence of RAE in IRD genes. This study details the molecular and biological pathways that may be enriched for both BAE and RAE in IRD genes. We propose that RAE may provide a novel mechanism to explain the phenotypic characteristics of IRDs. Future work should characterize allele expression profiles for IRD genes using ocular tissue. By identifying the embryologic timing of expression profile imprinting for IRD genes, we may be able to better explain the phenotypes observed in IRDs. Through an understanding of the genetic and molecular mechanisms that drive RAE and their consequences, we may gain new insights to therapeutics for these diseases. In conclusion, certain IRD genes appear to be under random allelic expression, which may have an influence on the resulting retinal phenotype.

**Supplementary Materials:** The following supporting information can be downloaded at: https://www.mdpi.com/article/10.3390/cimb45120625/s1, Table S1: List of genes from RetNet that exhibit biallelic expression (BAE); Table S2: Additional biologic processes significantly enriched for BAE; Table S3: GO ontology output tables for RAE and BAE.

**Author Contributions:** Conceptualization, C.J.R. and J.S.P.; methodology, C.J.R. and J.S.P.; software, C.J.R. and J.S.P.; validation, C.J.R. and J.S.P.; formal analysis, C.J.R.; investigation, C.J.R. and J.S.P.; resources, C.J.R. and J.S.P.; data curation, C.J.R. and J.S.P.; writing—original draft preparation, C.J.R.; writing—review and editing, C.J.R. and J.S.P.; visualization, C.J.R.; supervision, J.S.P.; project administration, J.S.P.; funding acquisition, J.S.P. All authors have read and agreed to the published version of the manuscript.

**Funding:** This work was supported in part by an unrestricted grant from the J. Arch McNamara, Research Fund for Wills Eye Hospital and Mid Atlantic Retina.

**Data Availability Statement:** The data are available in a publicly accessible repository. The data presented in this study are openly available at: https://doi.org/10.1136/bmjophth-2022-001079, Table 1; https://doi.org/10.1016/j.celrep.2022.111945, Tables S2 and S3; and GO enrichment analysis: https://geneontology.org/ (accessed 7 July 2023).

**Conflicts of Interest:** The authors declare no conflict of interest.

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
