# Peer review of "Random Allelic Expression in Inherited Retinal Disease Genes"

_cimb, doi:10.3390/cimb45120625_

Round 1
Reviewer 1 Report
Comments and Suggestions for Authors
This study by Richards and Pulido makes a significant contribution to the understanding of inherited retinal diseases (IRDs). Their approach to identifying random allelic expression (RAE) in IRD-causing genes is methodical and thorough, as evidenced by the cross-referencing of genes from the Retinal Information Network with recent literature. This method has successfully identified 31 out of 184 IRD-causing genes that exhibit RAE, providing new insights into the genetic mechanisms underlying these conditions.
The paper's exploration of loss-of-function intolerance (LOFI) in RAE and biallelic expression (BAE) genes offers a nuanced view of gene expression patterns in IRDs. The findings that a significant portion of brain tissue exhibited BAE in both sexes, and the identification of molecular and biological pathways significantly enriched among these genes, are particularly intriguing. These results could lead to a better understanding of the cellular mechanisms involved in IRDs and potentially open new avenues for treatment strategies.
Furthermore, the paper's discussion of allele-specific expression as a potential mechanism underlying the phenotypic variability of IRDs adds an important dimension to our understanding of these diseases. The suggestion that brain tissue preferentially exhibits BAE, possibly due to selective pressures against RAE, is a novel insight that merits further exploration.
Some minor improvements should be made.
18. Otto, S.P.; Goldstein, D.B. Recombination and the evolution of diploidy.. Genetics 1992, 131, 745–751. https://doi.org/10.1093/ge- 265
two peiods here.
The table colors are strange.
What are possible selection adavantages of BAE? This should be addressed more clearly. Why the RAE should be against, consider XCI is so widespread and adpative ?
XCI could be due to the stronger sexual selection in male, without affecting female too much.
For vision, I don't think sexual selection is so strong. Thus, BAE could be preferred.
Thus, the authors should not expect the same logic for sexual and adaptive selection. The observations in this study could be explained in adaptive trait context. Thus, BAE makes sense.
Author Response
Reviewer 1
This study by Richards and Pulido makes a significant contribution to the understanding of inherited retinal diseases (IRDs). Their approach to identifying random allelic expression (RAE) in IRD-causing genes is methodical and thorough, as evidenced by the cross-referencing of genes from the Retinal Information Network with recent literature. This method has successfully identified 31 out of 184 IRD-causing genes that exhibit RAE, providing new insights into the genetic mechanisms underlying these conditions.
The paper's exploration of loss-of-function intolerance (LOFI) in RAE and biallelic expression (BAE) genes offers a nuanced view of gene expression patterns in IRDs. The findings that a significant portion of brain tissue exhibited BAE in both sexes, and the identification of molecular and biological pathways significantly enriched among these genes, are particularly intriguing. These results could lead to a better understanding of the cellular mechanisms involved in IRDs and potentially open new avenues for treatment strategies.
Furthermore, the paper's discussion of allele-specific expression as a potential mechanism underlying the phenotypic variability of IRDs adds an important dimension to our understanding of these diseases. The suggestion that brain tissue preferentially exhibits BAE, possibly due to selective pressures against RAE, is a novel insight that merits further exploration.
Some minor improvements should be made.
- Otto, S.P.; Goldstein, D.B. Recombination and the evolution of diploidy.. Genetics 1992, 131, 745–751. https://doi.org/10.1093/ge- 265
two peiods here.
- Corrected, thank you
The table colors are strange.
- Updated
What are possible selection advantages of BAE? This should be addressed more clearly. Why the RAE should be against, consider XCI is so widespread and adaptive? XCI could be due to the stronger sexual selection in male, without affecting female too much. For vision, I don't think sexual selection is so strong. Thus, BAE could be preferred. Thus, the authors should not expect the same logic for sexual and adaptive selection. The observations in this study could be explained in adaptive trait context. Thus, BAE makes sense.
- Thank you, this brings up an excellent point.
- Selection advantages of BAE:
- Genes enriched for housekeeping functions have a tendency to have higher expression levels, as do genes under BAE (Reinius et al, 2015)
- Enriched among genes involved in essential nuclear functions (DNA organization, gene regulation, cell cycle). Enriched within pathways for essential genes that would prove lethal if a mutation arises. Preserved evolutionarily. (Kravitz et al, 2023)
- Selection advantages of RAE (Gendrel et al, 2016):
- Enhance specificity of the response to antigens or sensory stimuli (olfactory receptors, Purkinje neurons during brain development, protocadherins implicated in self-recognition processes (RAE could reduce neurons making self-contacts during this process))
- Ensure adequate gene dosage (similar to XCI)
- Fine-tune expression in different proportions of genes, for processes that are extremely dosage dependent (e.g Eya1 and otic development)
- Stage and tissue-specific, general increase in RAE during differentiation
- Important for brain development, neuronal excitability and plasticity
- Somatic mosaicism, could produce virtually infinite cellular diversity
- We have added the following (188-198):
- “Prior work has shown BAE is enriched in genes for essential nuclear functions, muta-tion-intolerant genes, and is preserved evolutionarily[7], [14]. Selection against RAE in brain tissue versus body tissue may reflect a difference in mutation tolerance for these tissues (i.e. body tissues may better tolerate a mutation that would otherwise portend a genetic disadvantage if occurring in brain tissue). Also, an evolutionary drive to preserve critical neurological pathways essential for survival may account for decreased RAE in brain tissues. Within XCI, significant intra-tissue and inter-tissue variability exists within individuals[22]. RAE may exhibit similar variability, a phenomenon that would not be captured in our evaluation of brain tissue from various regions. A potential outcome of this would be diseases with variable expressivity and diseases that display local, ocular phenotypes, without a systemic association.”
Reviewer 2 Report
Comments and Suggestions for Authors
Dear Author(s),
Thank you for sending me this paper entitled “Random Allelic Expression in Inherited Retinal Dystrophy Genes”, which I was pleased to consider for revision and enjoyed reading. This is a brief report which define the role of allelic expression among different Inherited retinal diseases.
Allele-specific expression is an important component of phenotypic variability observed in genetic diseases and this are specifically true for the IRDs. A comprehensive grasp of the mechanisms associated with gene expression is essential in various Inherited Retinal Disorders (IRDs). It will contribute to a more thorough explanation of the substantial phenotypic variability observed in different IRDs that stem from identical genotypes.
I find this paper interesting to read, with very few concerns from my side:
1. The introduction section provides relevant information. Few advice:
- Please rephrase the sentence in line 40-41 “assumed that there is equal intracellular expression intracellularly from…”
-Just clearly specify which “other publicly available genetic resources” have been considered to define the role of RAE and BAE.
2. Regarding the material and methods: The study design is clear and the method section sufficiently detailed. A minor suggestion: In line 97, 104,106 the authors reported a p value (p < 0.001; p = 0.257, p = 0.193; what this p is referred to? Which statistical test has been used ? Were p-values and/or confidence intervals reported? Please, specify it in the methods.
3. Result section: The results and the findings unambiguously address the purpose of the paper.
- Just a consideration: the author reported the findings of gene MFN2 in Table 1. The mutation in this specific gene account for an acute optic neuropathy associated with Charcot Marie Tooth disease whereas the other genes are all associated with different form of IRDs. Thus, consider deleting it from the table or explain the reason why the authors have included this phenotype.
This is the same for OPA1 gene in Table 3.
4.Tables and figures stand on their own; in my point of view, but this is just my preference, I do prefer a different color for the table which better highlight what stated inside.
At the end of the discussion, as for the limitations It would be more appropriate to highlight the strong points of the study.
To conclude, I think that the research question is overall original and relevant to the reader of the journal.
I look forward to hearing from you in due time regarding our submission and to respond to any further questions and comments you may have.
Author Response
All line numbers are in reference to those with "All Changes" turned on in tracking changes mode.
Dear Author(s),
Thank you for sending me this paper entitled “Random Allelic Expression in Inherited Retinal Dystrophy Genes”, which I was pleased to consider for revision and enjoyed reading. This is a brief report which define the role of allelic expression among different Inherited retinal diseases.
Allele-specific expression is an important component of phenotypic variability observed in genetic diseases and this are specifically true for the IRDs. A comprehensive grasp of the mechanisms associated with gene expression is essential in various Inherited Retinal Disorders (IRDs). It will contribute to a more thorough explanation of the substantial phenotypic variability observed in different IRDs that stem from identical genotypes.
I find this paper interesting to read, with very few concerns from my side:
- The introduction section provides relevant information. Few advice:
- Please rephrase the sentence in line 40-41 “assumed that there is equal intracellular expression intracellularly from…”
- Thank you, rephrased to read “ ..assumed that there is equal intracellular expression of maternal and paternal alleles…”, see lines 42-43.
-Just clearly specify which “other publicly available genetic resources” have been considered to define the role of RAE and BAE.
- Updated this sentence to include citations for the Tanner, 2022 dataset and GO knowledgebase. See lines 53-54.
- Regarding the material and methods: The study design is clear and the method section sufficiently detailed. A minor suggestion: In line 97, 104,106 the authors reported a p value (p < 0.001; p = 0.257, p = 0.193; what this p is referred to? Which statistical test has been used ? Were p-values and/or confidence intervals reported? Please, specify it in the methods.
- Thank you for the suggestion, the sentences have been reworded to specify what the p-value is in reference to (see lines 121-124 and 111-112)
- We have added lines 78-80 to the end of the methods to describe which statistical tests were used.
- Result section: The results and the findings unambiguously address the purpose of the paper.
- Just a consideration: the author reported the findings of gene MFN2 in Table 1. The mutation in this specific gene account for an acute optic neuropathy associated with Charcot Marie Tooth disease whereas the other genes are all associated with different form of IRDs. Thus, consider deleting it from the table or explain the reason why the authors have included this phenotype.
This is the same for OPA1 gene in Table 3.
- Thank you for this suggestion. We have included an explanation in the methods, lines 57-62, for why these genes are included in our analysis of IRDs. The pathogenesis of optic atrophy in these diseases (MFN2 and OPA1) are thought to arise from a primary degeneration of retinal ganglion cells, and would therefore be considered an IRD.
4.Tables and figures stand on their own; in my point of view, but this is just my preference, I do prefer a different color for the table which better highlight what stated inside.
- Updated
At the end of the discussion, as for the limitations It would be more appropriate to highlight the strong points of the study.
- Thank you, please see lines 237-246 where we have added a paragraph detailing the strong points of this study and the future work that must be conducted in this space.
To conclude, I think that the research question is overall original and relevant to the reader of the journal.
Reviewer 3 Report
Comments and Suggestions for Authors
Reviewer comments for Richards & Pulido. Random Allelic Expression in Inherited Retinal Dystrophy Genes
The authors perform a secondary analysis of Kravitz et al and RetNet data to investigate random versus biallelic expression as an influence on IRD phenotypes. They demonstrate a comprehensive understanding of the pathogenesis of these diseases, and this is a worthy addition to the literature on IRD pathogenesis. I have only minor comments. Please see below.
Comments:
Abstract: You use ‘dystrophy’ in the title and ‘diseases’ in the abstract then back to ‘dystrophy’ in the results and discussion. ‘Degenerations’ is another option. Consistently use one throughout the manuscript.
‘IRDs are responsible for visual loss…’ is an overly simplistic statement. Although these are critical diseases which remain an unmet need, they typically affect children and/or the working age population, so an adjustment would help to inform the reader of their place within the causes of blindness e.g. ‘IRDs are a significant contributor to visual loss in children and young adults, falling second only to diabetic retinopathy (England/Wales/Australia).
Please spell out abbreviations at first use e.g., GO ontology line 14
Intro:
Line 38: suggest changing to macular dystrophy rather than degeneration.
Line 39: suggest use terminology ‘syndromic’ alongside ‘extraocular.’
Materials/Methods:
Lines 53-54: this sentence is not grammatically correct. Perhaps adjust to ‘Genes listed in the RIN were assessed using only mapped and identified genes for further analysis.’
Line 68: spell out abbreviation for GO (looks like ‘gene ontology’ so the abbreviation should just be GO, not ‘GO ontology.’
Results:
Lines 71-76: These sentences were already stated in the methods, lines 54-56. I would start your results with the 282 genes line.
Typically, do not start a sentence with a number, but spell it out.
Lines 80-82: Were there no genes where expression varied between tissues i.e., BAE in brain and RAE in skin etc. I see a comment on this on lines 95-97, but please elaborate.
P values: although these are highly significant, standardization helps interpretation, so I would suggest adjusting to p<0.001 unless there is a valid statistical reason for exactness.
Discussion:
Line 150: use XCI as previously
Conclusion:
These sentences should be merged as follows: ‘In conclusion, certain IRD genes appear to exhibit random allelic expression, which may have an influence on the resulting retinal phenotype.’
Comments on the Quality of English LanguageMinor grammatical errors, see comments to authors. Generally grammatically correct.
Author Response
The authors perform a secondary analysis of Kravitz et al and RetNet data to investigate random versus biallelic expression as an influence on IRD phenotypes. They demonstrate a comprehensive understanding of the pathogenesis of these diseases, and this is a worthy addition to the literature on IRD pathogenesis. I have only minor comments. Please see below.
Comments:
Abstract: You use ‘dystrophy’ in the title and ‘diseases’ in the abstract then back to ‘dystrophy’ in the results and discussion. ‘Degenerations’ is another option. Consistently use one throughout the manuscript.
- Thank you for this observation, we have changed all references (outside of disease names or direct quotes) to be “Inherited Retinal Diseases”, which is the more common terminology for this group of disorders.
‘IRDs are responsible for visual loss…’ is an overly simplistic statement. Although these are critical diseases which remain an unmet need, they typically affect children and/or the working age population, so an adjustment would help to inform the reader of their place within the causes of blindness e.g. ‘IRDs are a significant contributor to visual loss in children and young adults, falling second only to diabetic retinopathy (England/Wales/Australia).
- Thank you for this suggestion, we have updated the abstract (see lines 8-10).
Please spell out abbreviations at first use e.g., GO ontology line 14
- Thank you, see line 15, we have changed this line to the more generic term “gene ontology”, without reference to the GO Knowledgebase. We have updated line 54 in the introduction to reference the GO Knowledgebase.
Intro:
Line 38: suggest changing to macular dystrophy rather than degeneration.
- Thank you, see line 40 for updated wording.
Line 39: suggest use terminology ‘syndromic’ alongside ‘extraocular.’
- Thank you, see line 41-42 for the updated wording.
Materials/Methods:
Lines 53-54: this sentence is not grammatically correct. Perhaps adjust to ‘Genes listed in the RIN were assessed using only mapped and identified genes for further analysis.’
- Thank you, please see lines 57-58 for the updated wording.
Line 68: spell out abbreviation for GO (looks like ‘gene ontology’ so the abbreviation should just be GO, not ‘GO ontology.’
- Thank you, please see lines 76-77 for the updated wording. Line 54 includes the first reference to the Gene Ontology (GO) Knowledgebase.
Results:
Lines 71-76: These sentences were already stated in the methods, lines 54-56. I would start your results with the 282 genes line.
- Thank you, we have removed the requisite sentences, see lines 82-87.
Typically, do not start a sentence with a number, but spell it out.
- Thank you, we have updated lines 87, 90, 92.
Lines 80-82: Were there no genes where expression varied between tissues i.e., BAE in brain and RAE in skin etc. I see a comment on this on lines 95-97, but please elaborate.
- Thank you for this suggestion. Table 2 lists the genes for which there is a discordant expression profile between all tissues and brain tissues. The implications of all tissues having significantly more genes under RAE versus brain tissue are discussed in lines 188-198.
P values: although these are highly significant, standardization helps interpretation, so I would suggest adjusting to p<0.001 unless there is a valid statistical reason for exactness.
- Thank you, we have updated lines 129-142 to reflect this suggestion. We have also updated the lines 129-130 to spell out the acronym for False Detection Rate (FDR) at first use.
Discussion:
Line 150: use XCI as previously
- Thank you, we have updated lines 170 and 222, replacing X-chromosome inactivation with XCI.
Conclusion:
These sentences should be merged as follows: ‘In conclusion, certain IRD genes appear to exhibit random allelic expression, which may have an influence on the resulting retinal phenotype.’
- Thank you, we have updated line 235.